# Evidence of the Beneficial Effects of Ursolic Acid against Lung Cancer

**DOI:** 10.3390/molecules27217466

**Published:** 2022-11-02

**Authors:** Amanda Kornel, Matteo Nadile, Evangelia Tsiani

**Affiliations:** 1Department of Health Sciences, Faculty of Applied Health Sciences, Brock University, St. Catharines, ON L2S 3A1, Canada; 2Centre for Bone and Muscle Health, Brock University, St. Catharines, ON L2S 3A1, Canada

**Keywords:** lung cancer, ursolic acid, proliferation, survival, invasion, metastasis, signaling cascades

## Abstract

Lung cancer is the leading cause of cancer-related deaths globally. Despite current treatment approaches that include surgery, chemotherapy, radiation and immunotherapies, lung cancer accounted for 1.79 million deaths worldwide in 2020, emphasizing the urgent need to find novel agents and approaches for more effective treatment. Traditionally, chemicals derived from plants, such as paclitaxel and docetaxel, have been used in cancer treatment, and in recent years, research has focused on finding other plant-derived chemicals that can be used in the fight against lung cancer. Ursolic acid is a polyphenol found in high concentrations in cranberries and other fruits and has been demonstrated to have anti-inflammatory, antioxidant and anticancer properties. In this review, we summarize recent research examining the effects of ursolic acid and its derivatives on lung cancer. Data from in vitro cell culture and in vivo animal studies show potent anticancer effects of ursolic acid and indicate the need for clinical studies.

## 1. Introduction

Cancer is characterized by enhanced proliferative signaling and its ability to resist programed cell death/apoptosis, evade growth suppressors and induce angiogenesis, leading to survival, growth, tumor formation and metastasis [1]. Lung cancer is the leading cause of cancer-related deaths globally [2,3,4] and is subdivided into small cell lung cancer (SCLC) [5] and non-small cell lung cancer (NSCLC) [6]. SCLC accounts for approximately 15% of global lung cancer cases, is highly aggressive and has a 7% five-year survival rate [5]. NSCLC accounts for 85% of all lung cancer cases; it is aggressive, with a five-year survival rate of approximately 25% [6].

Excessive growth factor receptor signaling is well established as a significant contributor driving cancer cell survival, proliferation and metastasis [7]. For example, certain mutations of epidermal growth factor receptor (EGFR) extracellular and kinase domains lead to downstream overactivation of signaling pathways, such as Ras–mitogen activated protein kinase (MAPK) and phosphatidylinositol 3-kinase (PI3K)/protein kinase B (Akt), which play a major role in cell proliferation and survival [8,9]. Activation of the PI3K/Akt pathway leads to downstream activation of the mammalian target of rapamycin (mTOR) and the ribosomal S6 kinase (p70S6K), resulting in increased protein synthesis, proliferation and inhibition of apoptosis, resulting in enhanced survival [9,10,11,12,13]. Mutations or overactivation of different key players of these signaling cascades, such as Ras and/or PI3K, not only result in cancer cell survival and enhanced proliferation but are responsible for the development of chemotherapy and radiotherapy resistance [14,15,16,17].

During stressful cellular events, such as DNA damage, tumor suppressor p53 initiates cell cycle arrest to allow for DNA repair, and when repair is not possible, cellular apoptosis occurs [18]. Mutations of tumor suppressor gene p53 result in loss of function and drive carcinogenesis [19,20].

Other signaling molecules that play a role in carcinogenesis include the nuclear factor kappa-light-chain enhancer of activated B cells (NF-kB). NF-kB acts as a transcription factor and is activated by tumor necrosis factor (TNF-a) and several ligands that bind to transmembrane protein receptors, such as growth factor receptors. NF-kB activation results in the activation/release of antiapoptotic factors, cell cycle regulators, cytokines and chemokines, leading to cell survival, proliferation, inflammation and angiogenesis [21]. Increased activation of NF-kB is associated with promotion of tumor cell proliferation and survival, whereas compounds that inhibit NF-kB have potential as cancer treatments [21].

Initiation of death receptor signaling activates the proapoptotic protein BID, which leads to the release of cytochrome C from mitochondria; the activation of effector caspases that cleave PARP; and the activation of other proteins responsible for cell shrinkage, DNA fragmentation and chromatin condensation, resulting in apoptosis [22,23], underscoring the importance of targeting the apoptotic cascades in certain cancer treatments [24].

Current treatments for lung cancer include surgery, chemotherapy, radiation therapy and immunotherapy [25]. Medications that counteract the mutations of EGFR signaling include tyrosine kinase inhibitors (TKIs) [26,27]. Other targeted therapies affecting tumor progression include PI3K pathway inhibitors, such as the FDA-approved orally administered GDC 0941, a potent pan-PI3K inhibitor, which prevents the progression and growth of tumors and the downstream activation of other proteins involved in cell growth and apoptosis [25].

Traditionally, plant-derived chemicals have been developed into agents used to treat cancer, including lung cancer. Such examples of plant-derived chemotherapy agents include paclitaxel and docetaxel, which are derived from the bark of the Pacific and European yew tree (*Taxus brevifolia* and *Taxus baccata*, respectively) [28,29]. Lung cancer often develops resistance to current therapy approaches [30,31], and the search for novel plant-derived chemicals that can counteract cancer and act as chemo- and/or radiation sensitizers is ongoing [32,33].

Ursolic acid (UA) is a pentacyclic triterpenoid (Figure 1) (chemical formula C_30_H_48_O_3_) [34] found in the leaves and fruits of more than 120 plant species [35,36,37,38,39,40,41,42,43,44,45,46,47,48] (Table 1). Many flowering plants contain UA, with high levels found in lavender, white deadnettle, marigold, rosinweed, basil, rosemary, daylily and olive tree leaves [49,50]. Fruits including black elderberries, apples, cranberries and pears [46] contain substantial levels of UA (Table 1), as does olive oil [48].

There is evidence that UA exhibits a wide range of biological activities [34,37,51,52], including anti-inflammatory [53,54], neuroprotective [55,56], antidiabetic [57], and anticancer properties [58,59,60,61,62].

A few published reviews summarize the effects of UA in cancer prevention and treatment [60,63,64], but none have focused on lung cancer. The focus of the current review is on the studies that examined the effects of UA against lung cancer. Studies performed both in vitro utilizing cell cultures and in vivo utilizing lung cancer animal models were reviewed and summarized and are presented chronologically in ascending order. In addition, the present review includes studies examining the effects of UA derivatives and how they compare to the parent compound. Importantly, our review focuses on the signaling molecules/cascades affected by UA treatment in an attempt to identify the mechanisms involved.

## 2. Effects of Ursolic Acid against Lung Cancer

### 2.1. Effects of Ursolic Acid against Lung Cancer: In Vitro Studies

Studies that have examined the effects of UA in cultured human lung cancer cells are summarized below and in Table 2.

Human A549 adenocarcinoma lung cancer cells treated with ursolic acid (2–40 µM) had significantly reduced proliferation, G1-phase cell cycle arrest and induction of apoptosis (determined via DNA fragmentation), (Table 2). Western blot analysis revealed that UA-treated A549 cells had increased p53 protein levels and decreased levels of cyclin D1, D2, E and NF-kB/p65 proteins. The levels of the antiapoptotic proteins B-cell lymphoma 2 (Bcl-2) and Bcl-X_L_ were significantly reduced, whereas Bax protein levels were increased. Treatment with UA induced p21/WAF1 and Fas/APO-1 signaling [65], (Table 2). Loss or reduced levels of the tumor suppressor p53 contributes to cancer initiation and progression [19,20], and the evidence from this study showing an increase in p53 levels with UA treatment is of major significance, indicating its tumor suppression potential. In addition, the study provided strong evidence of induction of lung cancer cell apoptosis with UA treatment, indicating its potential to be used in the treatment of lung cancer.

**Table 2 molecules-27-07466-t002:** Evidence of the effects of ursolic acid in lung cancer cells: a summary of in vitro studies.

Cell Type	Dose/Duration	Findings	Mechanism	Reference
A549	UA 2–40 µM 0–72 h	↓ Proliferation ↑ Apoptosis G1 phase cell cycle arrest	↑ p53 protein ↓ Cyclin D1, D2 and E ↑ Fas/APO-1 receptor ↑ FasL ↑ Bax protein ↓ NF-kB/p65 activity ↓ Bcl-2 protein ↓ Bcl-Xl protein	[65]
H460	UA 3, 10 and 30 µM 24 h	↑ Apoptosis ↓ Proliferation ↓ Migration	↑ Cleaved caspase-3 ↑ MMP 1, 2, 3, 9 and 10 gene expression ↑ Cytosolic glucocorticoid receptor	[66]
A549 H3255 Calu-6	UA 2, 4, 8 and 16 µM	↑ Apoptosis ↓ Cell viability ↓ Cell migration	↓ NA^+^-K^+^-ATPase activity ↓ PKC activity ↓ VEGF protein ↓ ICAM-1 mRNA ↓ Fibronectin mRNA ↓ MMP 2 and 9 mRNA	[67]
A549	UA 5–20 µM 24 h	↓ Proliferation ↓ Cell adhesion ↓ Wound healing ↓ Cell migration	↑ E-cadherin ↓ N-cadherin ↓ vimentin ↓ AEG-1 ↓ NF-kB	[69]
A549/H460	UA 30 µM 12, 24 and 48 h	↓ Cell viability ↓ Proliferation ↑ Apoptosis ↑ Chromatin condensation	↑ Cleaved caspase 3/9 ↓ Bcl-2 ↑ Bax ↑ p-AMPK ↓ p-mTOR ↓ ACC activity ↓ FASN l activity	[70]
A549	UA 25, 50, 100, 250 and 500 µM	↓ Cell viability	↓ VRK1 autophosphorylation ↓ VRK1 activity ↓ p-CREB ↓ p-His-H3 ↓ Cyclin D1 mRNA	[71]
PC9, H1299, A549, H1650, H358 and H1975	UA 5, 10, 20, 30, 40, 50 and 80 µM 24, 48 and 72 h	↓ Cell growth ↑ Apoptosis	↑ pSAPK/JNK ↓ SP1 protein ↓ DNMT1 protein ↓ EZH2 protein	[72]
H28, H2452 and MSTO-211H	UA 0–80µM 24–72 h	↑ Cytotoxicity ↓ Proliferation ↑ Sub-G1 population ↓ EMT	↑ Cleaved caspase-3 ↑ Cleaved PARP ↑ E-cadherin ↓ N-cadherin ↓ β-catenin ↓ p-GSK2α/β ↓ cyclin D1 ↓ p-AKT ↓ NF-kB	[73]
A549	UA 10–100 µM 24 h	↓ Proliferation ↑ Apoptosis ↑ S-phase cell cycle arrest ↑ Autophagy	↓ Bcl-2 protein ↑ Cleaved PARP ↑ LC3-II/LC3-I ratio ↑ p62	[75]
A549	UA 11, 22, 44 and 88 µM 24 and 48 h	↓ Cell viability ↑ Autophagy ↑ Mitophagy	↑ LC3-II/LC3-I ratio ↑ p62 protein ↑ PINK1 protein ↓ p-AKT ↓ p-mTOR ↑ Nrf2 protein ↑ ROS	[76]
A549	UA 5, 10 and 20 µM 24, 48 and 72 h	↓ Stemness ↓ Chemoresistance	↓ CD133 ↓ Oct-4 ↓ Notch3 ↓ Nanog ↓ Sox2	[78]
H1975 NSCLC with EGFR T790M mutation	UA 1, 5, 25, 50 and 100 µM	↓ Cell growth ↑ Apoptosis ↓ Cell motility	↓ CT45A2 mRNA ↓ TCF4 ↓ p-β-catenin @ Ser33/37/Thr41 ↑ p-GSK-3b @ Ser9	[77]
H1975	UA 0.001–0.1 µM	↓ EMT	↓ N-cadherin ↑ E-cadherin ↓ MMP-2 and -9 ↓ TGF-β1	[79]
NCI-H292	UA 3, 6, 9, 12 and 15 µM 24 and 48 h	↓ Cell viability ↑ Apoptosis ↑ Ca^2+^ production ↓ Mitochondrial membrane potential	↑ Cleaved caspase-7 ↑ Cleaved PARP ↑ Chromatin condensation ↑ Cytochrome c ↑ endo G ↑ AIF protein ↓ Bcl-2 protein ↓ BID protein	[80]
A549, H460, H1975, H1299 and H520 H82 and H446 LLC	UA 5–40 µM/ 48 h	↓ Proliferation ↑ Apoptosis ↑ Autophagy ↓ Cell Viability	↑ Cleaved PARP ↓ Bcl-2 protein ↑ LC3-II protein ↓ p-S6K @ T389 ↓ p-S6 @ S240-244 ↓ p-4E-BPI @ S65 ↓ p-AKT	[81]
A549 and H460	UA 10 and 20 µM	↓ Proliferation ↑ Apoptosis ↑ G0/G1 cell cycle arrest ↓ Angiogenesis ↓ Migration ↓ Invasion ↓ Tumorsphere formation	↓ p-EGFR ↓ VEGF ↓ MMP-2 ↓ PD-L1 ↓ CDK4 mRNA and protein ↓ CCND1 mRNA ↓ CCNE1 mRNA ↑ CDKN1A mRNA ↑ CDKN1B mRNA	[82]
Human bronchial epithelial cells exposed to cigarette smoke extract	UA 3, 6, 12 and 25 µM	↓ CSE-induced cytotoxicity	↑ Nrf2 activity	[68]
H1299	UA 50 and 80 µM 24 h	↓ Cell survival ↑ Radiosensitivity	↓ GSH (intracellular) ↓ HIF-1α protein	[74]

Human NSCLC H460 cells treated with 10 µM UA for 24 h exhibited increased apoptosis, characterized by activation of caspase-3 and DNA fragmentation, which was associated with a significant increase in gene expression of matrix metalloproteinases (MMPs) 1, 2, 3, 9 and 10, as well as an increase in glucocorticoid receptor (GR) cytosolic localization. These data suggest that members of the MMP family may be involved not only in invasion but also apoptosis of cancer cells and that cytosolic localization of GR may be an important factor in the upregulation of MMP during ursolic-acid-induced apoptosis of H460 cells [66].

Treatment of A549, H3255 and Calu-6 human lung cancer cells with UA decreased cell viability, migration and invasion and increased DNA fragmentation and apoptosis. Treatment with UA led to decreased protein expression of angiogenic factors vascular endothelial growth factor (VEGF) and transforming growth factor beta1 (TGF-beta1), both of which are biomarkers associated with cancer cell metastasis. In addition, mRNA levels of intercellular adhesion molecule-1 (ICAM-1) fibronectin and matrix metalloproteinases (MMP) 2 and 9, all important for cell adhesion, migration and invasion, were reduced. The levels of lactate dehydrogenase (LDH) in the medium, an indicator of plasma membrane damage, were increased, whereas Na^+^-K^+^-ATPase and PKC activities were reduced. Taken together, these data suggest that UA treatment increases apoptosis and decreases viability and migration of human lung cancer cells [67]. Although the reduction in the angiogenic factors VEGF and TGF-beta1 is a strong indicator of reduced migration, the study did not include any assays assessing cell migration/metastasis. The use of the scratch-wound and/or transwell cell invasion assays would have benefited the study. In addition, protein levels (not only mRNA levels) of ICAM-1, MMPs and fibronectin should have been examined.

Apart from examining the effects of UA on lung cancer cells, researchers have examined the potential of UA to counteract the effects of cigarette smoke in lung cells. Normal human bronchial epithelial (NHBE) cells were exposed to cigarette smoke extract (CSE) to induce cytotoxicity and cell death in order to mimic the in vivo conditions of cigarette smoking. Treatment with UA resulted in reversal of many of the negative effects of CSE, including reduced CSE-induced cytotoxicity, reduced CSE-induced lactate dehydrogenase efflux and recovery of CSE-induced glutathione loss. CSE activated the Nrf2 pathway, as shown by the increased levels of NQO1, GST and Nrf2 proteins. Treatment with UA normalized these results to near-control levels and reduced the CSE-induced DNA damage. In this cell model, UA alleviated the cytotoxic effects of CSE and recovered the intracellular redox balance, suggesting that UA has a protective effect against CSE-induced cell injury. The authors concluded that UA has a potential role in both the prevention and treatment of lung tumors caused by cigarette smoking [68]. The model created by this research group provides preliminary evidence of UA acting against cigarette-smoke-induced changes in lung cells. However, this is an in vitro model, and further work utilizing animal models is required to determine whether UA protects against cigarette-smoke-induced damage. Strong evidence of a protective role of UA in animal models would provide a justification for clinical trials.

A549 cells treated with UA showed reduced proliferation, as well as reduced adhesion, wound healing and transwell migration, indicating inhibition of metastasis. Western blot analysis revealed an upregulation of E-cadherin and a downregulation of N-cadherin and vimentin, indicating an inhibition of epithelial–mesenchymal transition (EMT). Furthermore, decreased expression of astrocyte-elevated gene-1 (AEG-1) correlated with repression of NF-kB signaling was observed in cells treated with UA [69].

Treatment of A549 and H460 human lung cancer cells with UA significantly decreased cell viability and proliferation. UA increased chromatin condensation and induced apoptosis. Flow cytometry revealed an increase in cell cycle arrest in the G0/G1 phase. Western blot analysis revealed an increase in cleaved caspases 3 and 9, increased phosphorylation of the proapoptotic protein Bax and a decrease in phosphorylated Bcl-2, an anti-apoptotic protein, suggesting that UA induces apoptosis in these cells via a caspase-dependent pathway. Treatment with UA increased phosphorylation/activation of AMP-activated protein kinase (AMPK) and inhibited the phosphorylation/activation of mTOR and p70S6K proteins, which are involved in protein synthesis and cell growth. In addition, the levels and activity of the lipogenic enzymes acetyl-CoA carboxylase (ACC) and fatty acid synthase (FASN) were decreased due to increased AMPK activation [70]. These data clearly show that UA is a strong activator of the AMPK energy sensor. UA-induced AMPK activation was abolished by an siRNA approach, which downregulated LKB1. Activated AMPK leads to downstream inhibition of mTOR and p70S6K, resulting in inhibition of protein synthesis and, ultimately, inhibition of cell proliferation. The inhibition of lipogenesis is also the result of activation of AMPK, resulting in limited availability of lipids, which are important components for cell proliferation. Although the study did not provide any evidence, it is possible that the activation of AMPK is also responsible for the induction of apoptosis. An approach to knockout/downregulate AMPK would have provided stronger evidence of the role of AMPK in the UA-induced effects.

Treatment of A549 human lung cancer cells with ursolic acid (0–50 µM) resulted in reduced cell viability and increased DNA damage associated with inhibition of vaccinia-related kinase 1 (VRK1) autophosphorylation and reduced phosphorylation of its downstream substrates, CREB and histone H3. This UA treatment also decreased cyclin D1 mRNA levels, a downstream effector of CREB and histone H3. Nuclear magnetic resonance (NMR) titration experiments and in silico modeling showed that UA is able to bind directly to VRK1, causing inhibition of kinase activity. A combination of UA with doxorubicin resulted in greater reduction in cell viability compared to either compound alone, indicating a synergic effect. Furthermore, when UA was combined with the PARP-1 inhibitor veliparib, a significant reduction in cell viability was observed compared to UA alone, indicating that UA exerts synergistic effects when combined with drugs that target DNA damage repair genes [71]. The finding reported in this study, that UA binds directly to the catalytic domain of VRK1 inhibiting its kinase activity, is based on in silico modeling. Further work is required to confirm that UA binds to VRK1 in cells, and the use of VRK1 overexpression/knockdown (siRNA) approaches can provide evidence of the role of VRK1 in UA-induced apoptosis. The finding that UA enhanced the effect of doxorubicin and veliparib is significant, suggesting that UA has the potential to be used as an adjuvant chemotherapy agent against lung cancer.

Wu et al. reported significantly reduced cell viability and induction of apoptosis in NSCLC (H1299, H1650, A549, PC9 and H1975) cells treated with UA (5–80 µM). UA treatment increased phosphorylation/activation of stress-activated protein kinases/Jun amino-terminal kinases (SAPK/JNK) while simultaneously decreasing the protein levels of SP1, DNMT1 and EZH2. Detection of caspase 3/7 activity was increased by UA treatment, as observed with a Caspase 3/7 assay kit. Use of the SAPK/JNK inhibitor SP600125 abolished the effects of UA on SP1, DNMT1 and EZH2, confirming the involvement of this signaling pathway. Exogenous expression of SP1 or DNMT1 also abolished the UA-induced effects, providing strong evidence of their involvement. Based on these data, the authors concluded that UA inhibits NSCLC cell growth via SAPK/JNK-mediated inhibition of SP1, DNMT1 and EZH2 [72].

Treatment of mesothelioma cells (H28, H2452 and MSTO-211H) with 10–40 µM UA inhibited proliferation and colony formation in a dose-dependent manner. Cell-cycle analysis showed an accumulation in the sub-G1 phase compared to untreated controls, indicating induction of apoptosis. Western blot analysis showed reduced total PARP and reduced caspase 3, both of which are apoptosis-related proteins. UA treatment attenuated the levels of EMT-related proteins; E-cadherin expression was increased, whereas levels of vimentin, N-cadherin, twist, B-catenin and pGSK3a/b were decreased. The expression of the survival genes cyclin D1, p-AKT and NF-kB were also reduced with UA treatment. MicroRNA array and qRT-PCR results indicated that let7b was upregulated in H2452 and H28 cells treated with UA. Overall, the data reported in this study suggest that UA treatment can induce apoptosis in lung cancer cells through inhibition of EMT and increased expression of let7b [73]. Downregulation of let7b would have provided stronger evidence of its role in the UA-induced effects.

Song et al. (2017) examined the ability of UA to increase the radiosensitivity of H1299 NSCLC cells [74]. They established radioresistant H1299 NSCLC cells by expressing a mutant hypoxia inducible factor-1α (HIF-1α) and found that treatment with UA increased their radiosensitivity (as evidenced by reduced cell survival), and this effect was associated with reduced endogenous GSH and HIF-1α levels [74]. Although these findings suggest that UA has the potential to radiosensitize lung cancer cells in vitro, further studies are required to examine whether UA causes radiosensitization in animals xenografted with NSCLC cells. Once strong evidence is reported from in vivo animal studies, clinical studies will be required to examine whether UA causes radiosensitization in NSCL patients.

Treatment of A549 cells with UA (10–100 µM) for 24 h inhibited cell proliferation and induced apoptosis. UA decreased the number of cells in the G0/G1 phase in a dose-dependent manner and increased the number of cells in the S and G2/M phases, suggesting UA-induced cell cycle arrest in the S and G2/M phase. This UA treatment decreased protein expression of Bcl-2 and total PARP, increased levels of cleaved-PARP and increased markers for autophagy, including the LC3-II/LC3-I ratio and p62 protein expression [75].

Human lung cancer A549 cells treated with UA (10, 20 and 40 µg/mL) for 24 or 48 h showed reduced cell viability, as well as increased autophagy and mitophagy. Treatment with UA increased the levels of autophagy-associated protein LC3-II and increased the LC3-II/LC-I ratio. In addition, the mitochondria of cells treated with UA appeared to be fragmented, and the levels of p62 protein, which is required for mitophagy, were increased and colocalized with fragmented mitochondria. Furthermore, treatment with UA increased ROS production, increased Nrf2 and PTEN-induced kinase 1 (PINK1) protein levels and reduced phosphorylated Akt and mTOR levels [76]. The data clearly show inhibition of Akt and mTOR as a result of UA treatment and not activation, as stated in the abstract (possible typographical error). The data from the two above-mentioned studies [75,76] suggest induction of autophagy and mitophagy by UA treatment; however, whether autophagy and mitophagy are induced in lung tumors in vivo is not known.

Ursolic acid treatment of NSCLC H1975 cells with an EGFR T790M mutation, a major cause of epidermal growth factor receptor tyrosine kinase inhibitor (EGFR-TKI/erlotinib) resistance, resulted in inhibition of proliferation and motility, as well as induction of apoptosis. UA treatment resulted in inhibition of CT45A2 expression via β-catenin/TCF4 inhibition. These data show that UA may be a potential treatment for NSCLC with the EGFRl858R/T790M mutation [77]. This study included examination of the effects of UA in animals xenografted with lung cancer cells expressing the EGFR T790M mutation; the findings are reported below (see in vivo section).

Chen et al. constructed a paclitaxel resistance cell line, A549-PR, with increased expression of stemness markers CD133, Oct-4, Notch3, Nanog and Sox2. Treating these chemotherapy-resistant cells with UA resulted in decreased expression of these marker proteins, reduced aldehyde dehydrogenase 1 (ALDH1) activity and, importantly, increased sensitivity to paclitaxel. Treatment of A549-PR cells with UA resulted in reduced miR-149-5/MyD88 signaling. The UA-mediated inhibition of chemoresistance and stemness was partially reversed by overexpression of miR-149 and knockdown of MyD88 [78]. Paclitaxel is a chemotherapy agent commonly used in the treatment of lung cancer. Unfortunately, many patients treated with paclitaxel develop resistance; therefore, the identification of methods to overcome such resistance is highly desirable, as it will improve patient survival. The findings of this study are significant; however, they are derived from in vitro studies only. Further in vivo animal studies and human clinical trials are required to confirm whether UA has the ability to overcome paclitaxel resistance.

Human NSCLC cells (H1975) treated with UA exhibited inhibited mesenchymal-like responses, such as migration, invasion and matrix metallopeptidase (MMP) 2 and 9 activity [79]. Furthermore, treatment with UA attenuated the transforming growth factor-β1 (TGF-β1)-induced decrease in E-cadherin and increased N-cadherin levels. TGF-β1-induced αVβ5 integrin levels and cell migration and invasion (using scratch and transwell invasion assays) were attenuated by UA treatment. The use of a pharmacological inhibitor of integrin (SB273005), similarly to UA treatment, reduced TGF-β1-stimulated cell migration and invasion, whereas combined treatment with UA and the inhibitor did not have any synergic effect. Although these data indicate that UA may target/reduce αVβ5 levels, leading to reduced epithelial-to-mesenchymal transition in NSCLC [79], more robust evidence is required to confirm this phenomenon. The use of inhibitors in general may lead to inhibition of other off-target molecules, and the specificity of SB273005 against αVβ5 versus other integrins is not clear. In addition, it is not known whether UA can bind to αVβ5. Further studies are required to answer these questions.

Human lung cancer cells (NCI-H292) treated with UA showed significantly reduced viability and increased apoptosis, as evidenced by increased levels of cleaved PARP and increased cytochrome C levels. Western blotting analysis revealed increased levels of endonuclease G (Endo G) and apoptosis inducing factor (AIF), whereas levels of the antiapoptotic proteins B-cell lymphoma 2 (BCL2) and BH3 interacting domain death agonist (BID) were reduced with UA treatment. In these cells, UA increased intracellular calcium levels by causing a time-dependent release and reduced mitochondrial membrane potential without affecting ROS levels. Taken together, these results suggest that UA treatment can induce apoptosis through AIF and Endo G release via a mitochondria-dependent pathway [80].

A variety of lung cancer cells (NSCLC: A549, H1975, H1299, H460 and H520; SCLC: H82 and H446; murine Lewis lung carcinoma cell line, LLC), when treated with UA, exhibited significantly reduced cell proliferation, whereas apoptosis was induced, as evidenced by an annexin V-FITC/PI double staining assay. UA treatment increased levels of cleaved PARP and decreased levels of Bcl-2, both indicators of apoptosis. In addition, treatment of H460, H1975 and A549 lung cancer cells with UA induced autophagy, as revealed by the elevation in the protein levels of the autophagy marker LC-3-II. Furthermore, UA treatment decreased phosphorylation/activation of Akt and its downstream targets, p70S6K and 4EBP1. Inhibition of autophagy, using siRNA for autophagy-related gene 5 (ATG5) or chloroquine, enhanced the effects of UA (inhibition of proliferation and induction of apoptosis), suggesting that the autophagy observed following UA treatment acts as a pro-survival mechanism in these cells [81].

Human A549 and H460 lung cancer cells treated with UA exhibited reduced proliferation, as well as reduced CDK4, cyclin D1 and cyclin E protein and mRNA levels, whereas the levels of p21 and p27 tumor suppressor proteins were increased. In addition, treatment with UA resulted in decreased VEGFR and pSTAT3 levels, suggesting inhibition of angiogenesis. Decreases in cell invasion and wound healing/migration were observed, suggesting antimetastatic activity of UA in NSCLC cells. Cancer stem cell (CSC) proliferation was inhibited with UA treatment, as evidenced by a significant reduction in the levels of CSC markers (NANOG, POU5F1 and SOX2). Molecular binding studies showed a high-affinity interaction between UA and EGFR, suggesting that UA interferes with EGFR signaling. This was further tested by treating cells with human recombinant EGF combined with 20 µM of UA. In the presence of UA, the EGF-mediated EGFR phosphorylation was reduced, and the downstream targets of the EGFR pathway, JAK2 and STAT3, showed reduced phosphorylation, whereas the total protein levels remained unchanged. Chromatin DNA extracted from NSCLC cells that had been treated with UA exhibited a significant reduction in the binding of STAT3 to promoters MMP2 and PD-L1. Based on these results, the authors concluded that UA has anticancer activity dependent on STAT3 signaling, that STAT3 acts as a bridge between EGFR and PD-L1 and that UA may be a drug candidate for PD-L1-based targeted cancer therapies [82]. Although the authors suggested that UA acts as an inhibitor of EGFR, further studies are required to confirm this hypothesis. Future in vitro studies utilizing recombinant EGFR and UA can clarify this issue.

According to the results of the studies presented above, it is evident that the effects of UA were examined in a variety of lung cancer cells representing distinct subtypes of the disease. Small cell lung cancer (SCLC) and non-small cell lung cancer (NSCLC) adenocarcinoma, squamous cell carcinoma and large cell carcinoma cell lines were utilized. The effective concentration of UA appears to be in the range of 20–50 µM in the majority of the studies reviewed herein. Unfortunately, no studies have been conducted that examine and directly compare the UA concentration required for half-maximum inhibition (IC_50_ values) of proliferation of different lung cancer cell lines.

Overall, the in vitro studies presented above and in Table 2 show that treatment of lung cancer cells with UA results in inhibition of proliferation, induction of apoptosis and autophagy through inhibition of Akt, NF-kB and mTOR; increased LC3-II/LC-3-I ratio; and increased cleavage of caspase 3/7 and PARP (Figure 2).

### 2.2. Effects of Ursolic Acid against Lung Cancer: In Vivo Studies

A limited number of studies have examined the effects of UA administration in mice xenografted with lung cancer cells. All the available studies are presented below and in Table 3.

Nude mice were intragastrically administered UA (10 mg/kg/day) for 1 week, followed by subcutaneous injection of A549 lung cancer cells in their right flank in the second week. Four weeks following cancer cell injection, the excised tumors showed reduced volumes compared to the control group that received no UA pretreatment. The tumor volumes observed following UA pretreatment were comparable to those in mice treated with the established chemotherapy drug cyclophosphamide [68]. However, (i) it is not clear whether UA treatment was continued for the 4 weeks after the injection of the lung cancer cells, and (ii) the dose and duration of treatment with cyclophosphamide were not reported. Given that the authors compared the effects of UA to cyclophosphamide, we assume that the length of both treatments (UA and cyclophosphamide) was 4 weeks. However, based on the experimental design of the study (pretreatment of lung cancer cells with UA before injection), it is not clear whether the effects observed following UA treatment are the result of the 1-week pretreatment, the 4-week treatment after the cancer cell injection or both. An improved experimental design is required to clarify this issue.

C57 BL/6 mice injected with Lewis lung carcinoma (LLC)-luciferase cells (1 × 10^7^ cells/mouse) in their flank were treated intraperitoneally with UA (100 mg/kg), etoposide (6 mg/kg) or a combination of UA and etoposide for 14 days. Both UA and etoposide treatment showed a trend toward reduced tumor weight; however, tumor weights did not significantly differ from those of non-treated animals in the control group. When UA and etoposide were administered in combination, a significant decrease in tumor volume was observed, indicating a synergistic effect. UA was also injected in combination with doxorubicin (2 mg/kg) to the xenografted mice for 10 days; this combination showed synergistic effects, resulting in reduced tumor size compared to each treatment alone [71]. Although these data suggest a chemosensitizing effect of UA, more studies are required to understand the mechanism involved in chemosensitization.

Subcutaneous injection of 50 or 100 mg/kg UA (extracted from the leaves of wild loquat E. fragrans Champ) to female Balb/c nude mice xenografted with A549 human lung cancer cells resulted in reduced tumor volume and weight compared to the vehicle control group [83]. Animals were injected with A549 cells (2 × 10^6^/mouse) 7 days prior to UA treatment, which was administered every other day for two weeks. Animals were sacrificed on experimental day 23; immunohistochemistry of tumor tissue samples from the high-dose UA group showed decreased protein expression of Ki-67, MMP-2 and CD34, as well as increased expression of Bid. The decreased levels of Ki-67 indicate decreased proliferation of cancer cells, the increased levels of Bid show induction of apoptosis and the reduced CD34 levels indicate reduced angiogenesis. The reduced proliferation, reduced angiogenesis and induction of apoptosis may explain the reduced tumor volume and weight. Although the reduction in MMP-2 suggests reduced metastasis, researchers found that liver metastasis was not prevented by UA treatment. Furthermore, the nude mice used in the study were found to have detectable levels of functional immune cells (T and NK cells); therefore, it is possible that UA increased the cancer-killing ability of the intrinsic immune system of the animals. Further studies are required to examine the role of UA in the immune system of animals and humans with lung cancer [83].

Athymic nude mice were injected with H1975 human lung cancer cells (5 × 10^6^), expressing the EGFR T790M mutation which confers resistance to erlotinib (EGFR-TKI), in their flank and administered UA (25 mg/kg) via daily injection for 18 days. This treatment reduced tumor growth and weight when compared to the vehicle control group [77]. UA-treated mice showed increased levels of TUNEL-positive cells in their tumor tissue, indicating apoptosis. Histological studies indicated that the UA treatment did not have any toxic effects on the liver or kidney tissue, suggesting that it is a safe treatment option and may exert cancer-cell-specific effects [77]. The findings of these studies are of major significance, as they indicate a potential of UA to overcome TKI/erlotinib resistance. However, whether the same applies to human lung cancer patients remains to be explored.

Athymic BALB/c nude mice injected with A549-PR cells, a lung cancer xenograft model resistant to established chemotherapy drug paclitaxel, had a higher tumor weight compared to mice xenografted with A549 cells. However, exposure of A549-PR cells to UA for 72 h prior to injection resulted in a significant attenuation of tumorigenesis 12 days following implantation, suggesting that UA attenuates induced chemoresistance. In vitro studies performed with these cell lines (see above, Section 2.1) provided evidence that UA treatment in these cells suppressed the miR-149/MyD88 pathway. Based on these combined data, the authors concluded that UA may be effective as a neoadjuvant cancer treatment against lung cancer by attenuating the stemness and chemoresistance via the miR-149-5p/MyD88 signaling axis [78].

Overall, these limited in vivo animal studies indicate that intragastric, intraperitoneal or subcutaneous administration of UA in lung cancer xenografted mice resulted in significantly reduced tumor volume and weight (Figure 3). Future studies should focus on the mechanisms and signaling pathways involved in UA-induced tumor reduction.

### 2.3. Ursolic Acid Derivatives and Their Effect against Lung Cancer

Owing to it anticancer properties but limited bioavailability in its natural state, researchers are looking for ways to manipulate UA and increase its effectiveness. A number of analogs of UA have been produced, and their effects have been examined using lung cancer cells. All the available studies are presented below and in Table 4.

Kanali et al. examined the effects of 32 UA derivatives utilizing human A549 lung cancer cells and found that the analog 4-bromoanilamideursolic acid (UA-9) had the greatest anticancer effect. Accordingly, UA derivatives with electron-donating groups (UA-9) were suggested to have the most potent anticancer activity [84]. However, all of the examined analogs had higher molecular weights and more limited polarity than UA, indicating potentially low solubility and therefore difficulty in passing through cell membranes.

Ursolic acid-triazolyl derivatives with o-bromo, o-chloro or o-methoxy substitutions on the aromatic ring showed inhibition of cell growth of cancer cells, including A549 lung cancer cells [85]. These UA-derivative compounds displayed higher anti-cancer effects than the parent ursolic acid, suggesting these derivatives should be further examined as potential cancer treatment options [85].

A group of ursolic acid benzylidene derivatives were created through an oxidation/condensation procedure and tested against cultured cancer cells, including A549 lung cancer cells [86]. Each of the compounds exhibited greater cancer cell cytotoxicity than UA and less cytotoxicity to normal FR-2 lung epithelial cells. The most promising derivative created was a UA derivative with 2,5-dihydroxy substitution on the aromatic ring, named 3b. Although this compound was further tested against colon cancer cells (HCT-116) and shown to induce apoptosis [86], studies utilizing lung cancer cells are lacking.

A series of A-ring cleaved UA derivatives were prepared and evaluated in NSCLC cells (H460, H322 and H460LKB1^+/+^). UA with a cleaved A-ring and a secondary amide at C_3_ (compound 17) was found to be the most active in inducing apoptosis. Treatment of lung cancer cells with this UA derivative induced apoptosis, as evidenced by the increased cleaved caspase-8 and caspase-7 levels and the decrease in Bcl-2 protein levels. Increased levels of Beclin-1 and LC3A/B-II suggested an induction of autophagy with this UA-derivative treatment. Decreases in mTOR and p62 protein levels were also observed. Based on these findings, UA derivative compound 17 may be a potential candidate for lung cancer treatment and should be further researched [87].

Human lung cancer cell (A549 and H460) treatment with the UA derivative UA232 resulted in reduced cell proliferation and induction of apoptosis. These effects were more prominent with UA232 than treatment with the parent UA compound, whereas the cytotoxic effects on normal cells (HEK293T) were the same. UA232 treatment of lung cancer cells caused the cells to arrest in the G0/G1 phase of the cell cycle in association with downregulation of cyclin D1 and CDK4. This novel UA derivative significantly increased apoptosis and increased cleaved-PARP1 protein levels compared to standard UA treatment. There was no significant change in the levels of Bax, Bcl-2 or caspase-8, indicating that apoptosis was not induced through either the mitochondrial apoptosis pathway or the death receptor pathway. UA232 treatment increased expression of CHOP, indicating a mechanism involving the ER stress pathway. Pretreatment with 4-PBA, an ER stress inhibitor, attenuated the UA232-induced apoptosis, a further indication of the role of ER stress [88].

Another UA derivative with functionalized aniline or amide side chains was synthesized and used to treat lung adenocarcinoma NCI-H460 cells. Compound 5Y8 had the most potent antiproliferative activity, which was significantly higher than that associated with treatment with the UA parent compound. Molecular docking studies revealed that compound 5Y8 had a key interaction with the active site of NF-kB, blocking the activity and signaling pathway, which led to apoptosis. These findings indicate that the 5Y8 UA derivative has potential as a new class of NF-kB inhibitor for the treatment of lung cancer and may contribute to overcoming chemotherapy resistance [89].

Treatment of A549 lung cancer cells with a number of UA derivatives containing long-chain diamine moieties resulted in significant inhibition of viability, with IC_50_ values in a micromolar range (5.22–8.95 μM) [90]. The UA derivative compound 8C was the most potent (IC_50_: 5.22 μM) and caused G1 phase cell cycle arrest and increased caspase 3 cleavage, an indicator of apoptosis. In addition, cancer cell migration was inhibited, suggesting that this compound may help to prevent metastasis. Treatment with UA compound 8C inhibited the NF-kB signaling pathway, as evidenced by the significantly reduced levels of phosphorylated IKKα/β and IKBα and reduced NF-kB levels. Furthermore, molecular docking studies showed a key interaction between compound 8C and the active site of NF-kB, blocking its activity [90]. Activated NF-kB is associated with increased cancer cell proliferation and inhibition of apoptosis; therefore, NF-kB inhibitors have potential as anticancer agents. This study [90] shows clear evidence of UA compound 8C targeting/inhibiting NF-kB activity in lung cancer cells; therefore, its anticancer potential should be further explored.

Although the parent ursolic acid compound has been shown to be an effective anticancer agent, the above-described studies have shown that derivatives of the parent UA compound have enhanced potency, indicating that modification may be an effective approach to drug development.

UA has limited bioavailability [50,91] due, in part, to its low water solubility and high molecular weight of 456.7 g/mol. A limited number of studies have examined UA bioavailability. Mice fed a diet containing 0.05% UA for 8 weeks were euthanized, and plasma UA levels and tissue distribution were measured utilizing high-performance liquid chromatography–mass spectrometry (HPLC-MS) [50]. Plasma UA levels of 580 ng/mL (1.26 µM) were observed, with the highest distribution in the liver (9.7 µg/g), followed by the colon (6.4 µg/g), kidney (5.9 µg/g), heart (3.9 µg/g), bladder (2.9 µg/g) and brain (1.6 µg/g). These data indicate that after oral administration of UA, significant plasma levels (micromolar) and tissue distribution can be achieved. In another study, Yang et al. prepared UA nanoparticles and measured plasma levels in rats after oral administration. One hour after rats were orally administered the parent UA or the nanoparticles at a dosage of 100 mg/kg of body weight, the resulting plasma levels were approximately 300 ng/mL (656.89 nM) and 1200 ng/mL, respectively. Plasma concentration dropped within 4 h in both groups of animals and remained at approximately 100 ng/mL (218.96 nM) for the next 12 h [91]. This study clearly indicates that although UA nanoparticles have increased gastrointestinal absorption, the parent UA compound is sufficiently absorbed to reach significant plasma levels (nanomolar). In another study UA or UA-phospholipid complex was administered at a dose of 10 mg/kg body weight to male rats, and the same trend was observed: an increase in plasma concentration to approximately 60 ng/mL (131.38 nM) and 125 ng/mL, respectively, after one hour, which rapidly reduced to approximately 10 ng/mL (21.90 nM) by hour five and remained at that level for the remining 24 h of the experiment [92]. However, no studies have been conducted examining UA plasma levels in humans after oral administration. In one study, subjects administered intravenous infusion of UA nanoliposomes (98 mg/m^2^) had increased plasma concentration with the peak UA concentration of 3404.6 ng/mL (7454.78 nM) four hours post infusion, followed by a rapid decline between hours 4 and 6. The concentration slowly declined from post-infusion hour six (approximately 300 ng/mL (656.89 nM)) to hour sixteen (approx. 30 ng/mL (65.69 nM)) [93].

Collectively, the limited animal studies indicate that oral administration of UA can result in plasma UA levels in the nano-to-micromolar range, at concentrations close to those used in the majority of the in vitro studies showing potent anticancer effects.

Another point that deserves consideration is the role of the gut microbiome in UA-induced effects. The gut microbiota may influence UA metabolism and result in the generation of UA metabolites with potent anticancer properties. Unfortunately, the anticancer properties of UA metabolites have not been studied to date.

## 3. Patent Applications and Clinical Trials Related to Ursolic Acid Use

A search of patent applications (using GooglePatents.com, accessed on 9 September 2022) revealed that in the past 10 years, 25,638 patent applications have been filed globally pertaining to UA, of which 5793 were related to UA use in cancer. In Canada, 354 applications were filed for UA patents, with 140 related to cancer. In the US, the numbers were higher, with a total of 1341 applications related to UA and 542 specifically related to UA and cancer. A search of European Union (EU) Clinical Trials Registry and the Government of Canada clinical trials registry yielded no registered trials related to ursolic acid use specifically for cancer treatment or any other use.

A search of ClinicalTrials.gov (accessed on 9 September 2022) related to ursolic acid revealed four trials: three completed and one withdrawn. Among the completed clinical trials, one (NCT02401113) examined the effect of ursolic acid (derived from loquat extract) in preventing sarcopenia in 54 adults, although no data/results have been published (trial completed Oct 2015). Another trial examined the bioavailability of ursolic acid in 18 healthy male adult participants (NCT04421716); although the study was completed in April 2021, no results have been published. The third trial (NCT02337933, trial completed Sept 2015) examined the effects of 12-week ursolic acid administration (150 mg administered orally once a day) in 24 adult participants with metabolic syndrome [94], with findings of reduced body weight, BMI, waist circumference and fasting blood glucose levels and improved insulin sensitivity. Approximately 50% of patients had transient remission of their metabolic syndrome. Patients with insulin resistance and metabolic syndrome are at an increased risk of developing cancer in general; although this clinical trial was not focused on cancer patients, the data are indirectly relevant to cancer. The reduced metabolic syndrome symptoms with UA use suggest improved metabolic control and potentially reduced cancer risk. One clinical trial (NCT04403568) involving the use of UA (150 mg, twice a day) for the treatment of prostate cancer was posted in May 2020. Unfortunately, this trial was withdrawn due to lack of funding.

These data clearly indicate that although there is evidence from in vitro studies and limited in vivo animal studies of the anticancer potential of ursolic acid, interest from the scientific community to perform clinical trials is limited, possibly due to a lack of strong in vivo animal studies, leading to a lack of funding.

One potential limitation of clinical translation of UA is its low bioavailability due to its low water solubility. An innovation that can combat this limitation would be to successfully encapsulate UA into micelles, nanoparticles or liposomes. These encapsulation strategies can increase the water solubility of UA, hopefully resolving the low bioavailability issue. In one preliminary clinical trial, the maximum tolerated dose (MTD), as well as the dose-limiting toxicity (DLT) of ursolic acid liposomes (11–130 mg/m^2^, administered by a 4 h intravenous infusion), was investigated in a group of 63 volunteer subjects. The DLT was found to be between 74–130 mg/m^2^ and mainly consisted of diarrhea and hepatotoxicity, and the MTD was determined to be 98 mg/m^2^ [95]. The above study [95] and the study by Xia et al. [93] described in Section 2.3, (intravenous infusion of UA nanoliposomes in humans) and the study by Yang et al. [91] described in Section 2.3 (oral administration of UA nanoparticles in rats) are the only studies published to date that attempted to examine dose, bioavailability and toxicity of UA nanoparticles. Furthermore, no studies have been conducted examining the effects of such UA nanoparticles against lung cancer (or any other cancer).

Although few UA derivatives have been found to be more effective than the parent compound in cell culture studies (studies presented in Section 2.3), none of them have been tested in animal or human studies. The few studies showing chemosensitizing [71,77,78] and radiosensitizing [74] properties of UA, have all been conducted in cell cultures. One potential future application of UA and UA derivatives is to be used as chemo- and/or radiosensitizing agents; therefore, we hope that in vivo studies utilizing lung cancer animal models will be performed in the future to evaluate such a potential.

## 4. Conclusions

In vitro studies reported here suggest that treatment of lung cancer cells with UA reduces proliferation and viability while increasing apoptosis and autophagy. Some studies showed cell cycle arrest in the G0/G1 phase, as well as reduced cell migration. Several studies showed inhibition of NF-kB and/or mTOR pathways, which are involved in cell proliferation and survival. In most of the studies examined herein, apoptosis was confirmed through increased levels of cleaved PARP, as well as caspases 3, 7 and 9. Some studies showed an increase in the expression of the proapoptotic protein Bax and a significant decrease in the expression of the antiapoptotic protein Bcl-2, highlighting the apoptotic effects of ursolic acid on lung cancer cells.

In vivo studies revealed that treatment of lung cancer xenografted animals with UA resulted in a decrease in tumor volume and weight; however, it is not yet clear whether the mechanisms involved are the same as those reported with cell cultures.

A number of UA derivatives and analogs have been developed and tested for their effect against lung cancer, with some showing higher anticancer effects than the parent UA compound. Further studies with derivatives are needed to examine the signaling pathways involved.

Drugs currently used for the treatment of lung cancer include cisplatin, gemcitabine, docetaxel, etoposide, paclitaxel and vinorelbine. Only a few studies have compared the effects of UA to the effects of currently used lung cancer drugs. Kim et al. [71] found that UA had similar effects as doxorubicin and veliparib in A549 lung cancer cells. Furthermore, when used in combination, UA enhanced the effect of doxorubicin and veliparib. In another study [78], UA was reported to overcome paclitaxel resistance in A549 cells.

In vivo animal studies have shown that UA treatment of animals xenografted with lung cancer cells had a similar effect in reducing tumor volume as cyclophosphamide [68], etoposide [71] and doxorubicin [71] and enhanced the effects of etoposide and doxorubicin when used in combination [71]. Although very limited, these studies provide strong evidence of the anticancer potential of UA.

Overall, the studies summarized in the present review collectively show that treatment of lung cancer cells with ursolic acid considerably reduces key cancer features, including cell viability, proliferation, colony formation and migration, in addition to inducing cancer cell death. Treatment of animal models of lung cancer with UA resulted in reduced tumor volume. Future research is required to further determine the effects of UA on both cancerous and normal tissues, as well as to elucidate the cellular signaling pathways involved.

It should be noted that UA has low water solubility and limited bioavailability. Future studies should be conducted to better understand the best route of administration, pharmacokinetics, bioavailability and tumor-reducing potential of UA, its derivatives and metabolites.

Importantly, to further investigate the anticancer potential of UA in cancer patients, clinical trials are necessary.

## Figures and Tables

**Figure 1 molecules-27-07466-f001:**
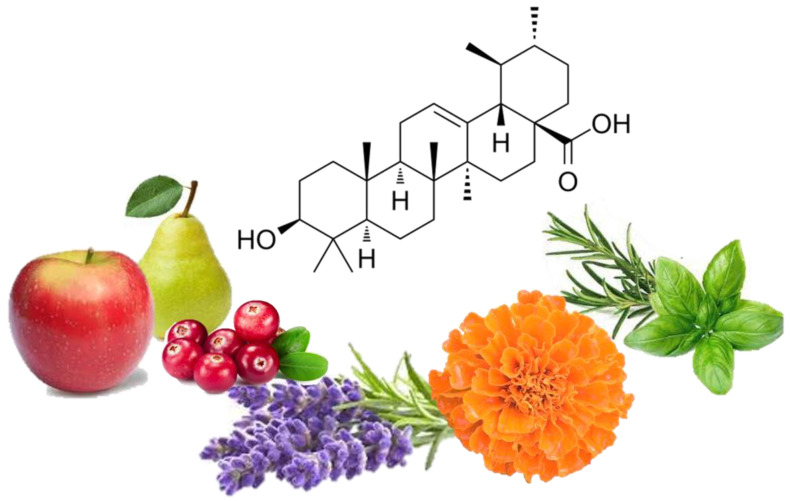
Chemical structure of ursolic acid, a compound present in the leaves and flowers of many plants, including lavender and marigold; fruits, such as apples, cranberries and pears; and herbs, including basil and rosemary.

**Figure 2 molecules-27-07466-f002:**
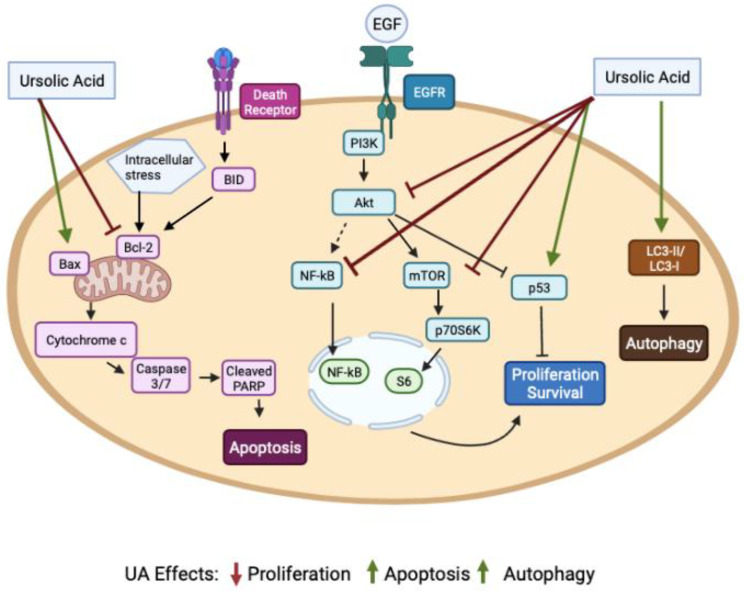
Ursolic acid reduced proliferation and induced apoptosis and autophagy of lung cancer cells in vitro. UA inhibited the phosphorylation/activation of Akt [73,81], mTOR [70,76] and NF-kB [65,73] and increased p53 protein levels [65], leading to inhibition of proliferation and survival. Furthermore, UA increased the proapoptotic protein Bax and decreased the antiapoptotic protein Bcl-2 [65,70,75]. An increase in cleaved caspases 3 and 7 was observed, leading to an increase in cleaved PARP and apoptosis [66,73,80]. An increase in LC3-II protein levels, a marker of autophagy, was also observed as a result of UA treatment [75,76,81]. The figure was created using BioRender.com (accessed on 9 September 2022) based on data from [65,66,69,70,73,75,76,80,81].

**Figure 3 molecules-27-07466-f003:**
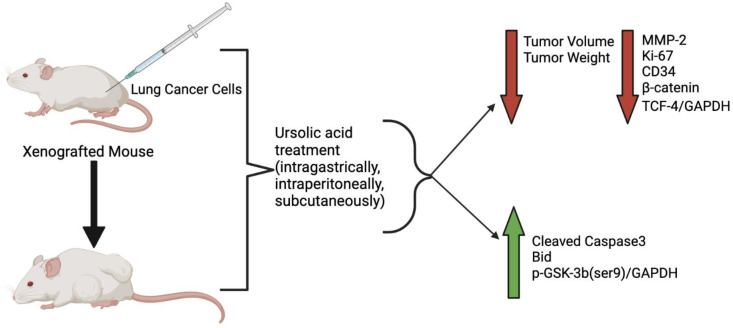
Effects of ursolic acid treatment on lung cancer in vivo. Mice xenografted with lung cancer cells (A549 or H1975), when treated with ursolic acid, had reduced tumor volume and weight compared to untreated animals. Ursolic acid treatment reduced the expression of MMP-2, Ki-67, CD34, β-catenin and TCF-4/GAPDH and increased cleaved caspase-3 and Bid protein levels. The figure, created using BioRender.com (accessed on 9 September 2022), is based on data from [68,71,77,78,83].

**Table 1 molecules-27-07466-t001:** Examples of ursolic acid concentrations in plants and fruits.

	Source		Concentration of UA	Reference
	Common Name	Botanical Name		
Plants	Lavender	*Lavandula*	106.7–153.1 mg/g 3.463–6.484 mg/g DW	[36]
	White deadnettle	*Lamii albi flos*	39.1–110.4 mg/g DW	[49]
	Marigold	*Calendula officinalis*	20.53 mg/g D.W	[38]
	Basil	*Ocimum tenuiflorum*	20.2 mg/g D.W	[39]
	Rosinweed, cup plant, compass plant	*Silphium sp. flowers*	17.95–22.05 mg/g D.W	[38]
	Rosemary	*Rosmarinus officinalis*	15.8–29.5 mg/g D.W	[40]
	Daylily	*Hemerocallis sp*	0.19 ± 0.05 mg/g DW	[50]
Fruits	Black elderberry extract	*Sambucus nigra L*	6.62 ± 0.26–0.002 mg/g	[42]
	Olive—adult olive tree leaves	*Olea europaea L.*	2.23 ± 0.1 mg/g	[41]
	Apple—apple peel	*Malus*	1.52 mg/g DW	[44]
	Apple—whole apple	*Malus*	0.77 ± 0.1 mg/g to 1.85 ± 0.17 mg/g	[43]
	Cranberry	*Vaccinium macrocarpon*	0.46–1.09 mg/g FW	[45]
	Pear—mature fruit peel	*Pyrus*	0.3481 mg/g	[47]
	Pear—young fruit	*Pyrus*	0.1293 mg/g FW	[46]
	Olive—virgin olive oil	*Olea europaea L.*	0.00138 ± 0.00015 mg/g	[48]

Abbreviations: DW: dry weight; FW: fresh weight.

**Table 3 molecules-27-07466-t003:** Effects of ursolic acid against lung cancer: in vivo evidence.

Xenograft Model	Dose/Duration	Findings	Mechanism	Reference
6–8-week nude mice A549 cells injected subcutaneously	UA—10 mg/kg intragastrical administration/ 1 week	↓ Tumor volume	Not investigated	[68]
C57 BL/6 mice injected with LLC-luciferase (1 × 10^7^ cells/mouse)	UA—100 mg/kg intraperitoneally injected	↓ Tumor volume ↓ Tumor weight	↓ VRK1 activity	[71]
Female Balb/c nude mice A549 cells 2 × 10^6^ cells/mouse	UA 50 or 100 mg/kg subcutaneous injection/every other day for 2 weeks	↓ Tumor growth ↓ Tumor weight	↓ MMP-2 ↓ Ki-67 ↓ CD34 ↑ Bid	[83]
Athymic nude mice H1975 cells subcutaneously injected 5 × 10^6^ cells/mouse	UA—25 mg/kg^−1^ daily for 18 days	↓ Tumor growth ↓ Tumor weight	Not investigated	[77]
Athymic Balb/c nude mice A549-PR cells	UA 20 µM 72hr pre-injection co-culture	↓ Tumorigenesis	Not investigated	[78]

**Table 4 molecules-27-07466-t004:** Ursolic acid derivatives and their effect against lung cancer in vitro. Chemical structures created with BioRender.com (accessed on 9 September 2022). Black portions indicate parent UA structure, and red segments indicate modifications.

Cell Line	Derivative Name	Derivative Structure	Findings	Mechanism	Reference
A549 SF-295 (CNS)	UA-9 10 nM	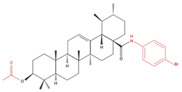	↓ Cell Density	Not investigated	[84]
A549	UA-triazolyl derivative	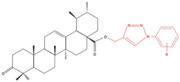	↓ Cell Density	Not investigated	[85]
A549	Compound 3B 50 µM, 48h	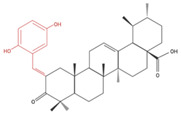	↓ Cell Growth	Not investigated	[86]
H460 H322	Compound 17	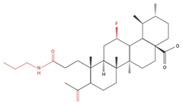	↑ Apoptosis ↑ Autophagy	↑ Cleaved caspases 8 and 7 ↑ Cleaved PARP ↑ LC3A/B-II ratio ↓ Bcl-2 protein ↓ mTOR protein	[87]
A549 H460	UA232 24, 48 and 72 h	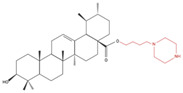	↑ Cell cycle arrest ↑ Apoptosis ↓ Proliferation	↓ Cyclin D1 protein ↓ CDK4 protein ↑ CHOP protein ↑ Cleaved PARP	[88]
NCI-H460	5Y8 5 and 10 µM	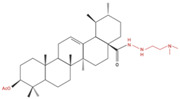	↑ Apoptosis G1 phase cell cycle arrest	↓ p-NF-kB ↓ p-IKKα/β ↓ TAK1 ↓ TAB1 ↑ ROS	[89]
A549	8c 5, 10 and 20 µM 24 h	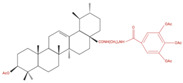	↑ Apoptosis G1 phase cell cycle arrest ↓ Cell migration	↑ Caspase-3 cleavage ↓ p-NF-kB ↓ p-IKBα ↓ p-IKKα/β	[90]

## Data Availability

Not Applicable.

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
