# Peer review of "Evidence of the Beneficial Effects of Ursolic Acid against Lung Cancer"

_molecules, 2022, doi:10.3390/molecules27217466_

Round 1
Reviewer 1 Report
This review “Evidence of the Beneficial Effects of Ursolic Acid Against Lung Cancer” by Kornel et al., provides an overview of several reports related the effects of ursolic acid on lung cancer. The topic is interesting, but there are key areas the authors need to address to strengthen the readability and critical viewpoints of this manuscript.
1. At the end of the introduction should be clearly stated what is the aim and the importance of this report beyond what other papers on the same topic have previously identified.
2. My main criticism is that section 2.1, 2.2 and 2.3 describe a list of reports related to the subtitle but they are not discussed. The information should be integrated, and authors should be able to highlight the central ideas of the references in order to critically discuss it. These sections need be appropriately edited.
3. The included tables in section 2.1, 2.2 and 2.3 are a hit for the review, but as long as the significance of the studies is not discussed, the information results very repetitive.
4. A better and complete explanation for figures should be included in figure legends.
5. The current section of conclusions contains the discussion of references that should be integrated into the sections 2.1, 2.2 or 2.3. The section of conclusions should contain information for perspectives or potential applications could be helpful for future research.
Author Response
Reviewer 1
Thank you for taking the time to review our manuscript. We followed your suggestions and addressed all your comments.
This review “Evidence of the Beneficial Effects of Ursolic Acid Against Lung Cancer” by Kornel et al., provides an overview of several reports related the effects of ursolic acid on lung cancer. The topic is interesting, but there are key areas the authors need to address to strengthen the readability and critical viewpoints of this manuscript.
- At the end of the introduction should be clearly stated what is the aim and the importance of this report beyond what other papers on the same topic have previously identified.
We have edited the introduction and in the revised manuscript the aim and the importance of our work/review are clearly stated.
- My main criticism is that section 2.1, 2.2 and 2.3 describe a list of reports related to the subtitle but they are not discussed. The information should be integrated, and authors should be able to highlight the central ideas of the references in order to critically discuss it. These sections need be appropriately edited.
We have edited sections 2.1, 2.2 and 2.3. The studies presented in these sections are discussed and the central ideas highlighted.
- The included tables in section 2.1, 2.2 and 2.3 are a hit for the review, but as long as the significance of the studies is not discussed, the information results very repetitive.
We enhanced the discussion of the significance of the studies presented in sections 2.1, 2.2 and 2.3.
- A better and complete explanation for figures should be included in figure legends.
The figure legends have been edited and include a clear explanation of the figure content.
Figure 2. Ursolic acid reduced proliferation while induced apoptosis and autophagy of lung cancer cells in vitro. UA inhibited the phosphorylation/activation of Akt [75,83], mTOR [78,85] and NF-kB [67,75] while increased p53 protein levels [67] leading to the inhibition of proliferation and survival. Furthermore, UA increased the proapoptotic protein Bax while decreased the anti-apoptotic protein Bcl-2 [67,77,85]. An increase in cleaved caspases -3, -7 was seen leading to an increase in cleaved PARP and apoptosis [68,75,82]. An increase in LC3-II protein levels, a marker of autophagy, was also seen with UA treatment [77,78,83]. The figure was created using BioRender.com, based on data from studies [67,68,71,75,77,78,82,83,85].
Figure 3. Effects of ursolic acid treatment on lung cancer in vivo. Mice xenografted with lung cancer cells (A549 or H1975), when treated with ursolic acid had reduced tumor volume and weight compared to non-treated animals. Ursolic acid treatment reduced the expression of MMP-2, Ki-67, CD34, b-catenin and TCF-4/GAPDH while increased cleaved caspase-3 and Bid protein levels. The figure, created using BioRender.com, is based on the data of the studies [70,73,79,80,86].
- The current section of conclusions contains the discussion of references that should be integrated into the sections 2.1, 2.2 or 2.3. The section of conclusions should contain information for perspectives or potential applications could be helpful for future research.
We followed the reviewer’s suggestion and have added information regarding the potential clinical translation and application of ursolic acid use in cancer. We added a new section titled “patent applications and clinical trials related to ursolic acid use”. This section is added before the conclusion section of the revised manuscript.
- Patent applications and clinical trials related to ursolic acid use
A search of the patent applications (using GooglePatents.com) revealed that, in the past 10 years, 25,638 patent applications have been filed globally pertaining to UA of which 5793 were related to UA use in cancer. In Canada there were 354 applications for UA patents with 140 related to cancer. In the US the numbers were higher with total 1341 applications related to UA and 542 specifically related to UA and cancer. A search of European Union (EU) Clinical Trials Registry and Government of Canada clinical trials registry found no trials registered related to ursolic acid use specifically for cancer treatment or any other use.
A search of ClinicalTrials.gov related to ursolic acid revealed four trials, 3 completed and 1 withdrawn. From the completed clinical trials one (NCT02401113) examined the effect of ursolic acid (derived from Loquat extract) on preventing sarcopenia in 54 adults but no data/results are posted (trial completed Oct 2015). Another trial examined the bioavailability of ursolic acid in 18 healthy male adult participants (NCT04421716) and although the study was completed in April 2021, no results are posted. The third trial (NCT02337933, trial completed Sept 2015) examined the effects of 12 week ursolic acid administration (150mg was given orally once a day) in 24 adult participants with metabolic syndrome [97] and found reduced body weight, BMI, waist circumference, reduced fasting blood glucose levels and improved insulin sensitivity. Around 50% of patients had transient remission of their metabolic syndrome. Patients with insulin resistance and metabolic syndrome have increased risk of developing cancer in general and although this clinical trial is not focused on cancer patients, the data are indirectly relevant to cancer. The reduced metabolic syndrome symptoms with UA use suggest better metabolic control and potential reduced cancer risk. One clinical trial (NCT04403568) involving use of UA (150 mg, twice a day) in the treatment of prostate cancer was posted in May 2020. Unfortunately, this trial was withdrawn due to lack of funding.
These data clearly indicate that although there is evidence from in vitro studies and limited in vivo animal studies of the anticancer potential of ursolic acid, interest from the scientific community, to perform clinical trials is very low. This may be due to lack of strong in vivo animal studies which leads to lack of funding.
One potential limitation of clinical translation of UA is its low bioavailability due to its low water solubility. An innovation that can combat this limitation would be to successfully encapsulate UA into micelles, nanoparticles or liposomes. These encapsulation strategies will increase its water solubility and hopefully will resolve the low bioavailability issue. In one preliminary clinical trial the maximum tolerated dose (MTD) as well as the dose-limiting toxicity (DLT) of ursolic acid liposomes (11-130 mg/m2, administered by a 4h intravenous infusion) was investigated in a group of 63 volunteer subjects. The DLT was found to be between 74 - 130 mg/m2 and mainly consisted of diarrhea and hepatotoxicity, and the MTD was determined to be 98 mg/m2 [98]. The above study [98] together with the study by Xia et al [96] described in section 2.3, (UA nano- liposomes intravenous infusion in humans) and the study by Yang et al [94], described in section 2.3, (UA nano-particles oral administration in rats) are the only studies up to now that attempted to examine dose, bioavailability and toxicity of UA nano-particles. Furthermore, there are no studies examining the effects of such UA nanoparticles against lung cancer (or any other cancer).
Although, few UA derivatives have been found to be more effective than the parent compound, in cell culture studies (studies presented in section 2.3) unfortunately, none of them have been tested in animal or human studies yet. The few studies showing chemo sensitization [73,79,80], and radio sensitization [76] properties of UA are all in cell cultures. One potential future application of UA and UA derivatives is to be used as chemo and/or radio sensitizing agents and hopefully, in vivo studies utilizing lung cancer animal models will be performed in the future to evaluate such a potential.
Reviewer 2 Report
This review covers the cytotoxicity and antitumor studies of UA against lung cancer cells. The information compiled in this review is comprehensive. However, the application and translation of the UA application are lacking in this review. For example, the efficacy of UA to different types of lung cancer was not evaluated/compared. Besides, patent and clinical trials related to UA were not included in this review too. Potential limitation and innovation to resolve or promote the clinical translation of the UA can be included too. The effect of UA comparing with the currently available lung cancer chemotherapeutic drug shall be evaluated.
Author Response
Reviewer 2
Thank you for taking the time to review our manuscript. We followed your suggestions and addressed all your comments.
Comments and Suggestions for Authors
This review covers the cytotoxicity and antitumor studies of UA against lung cancer cells. The information compiled in this review is comprehensive. However, the application and translation of the UA application are lacking in this review.
We followed the reviewer’s suggestion and have added information regarding the potential clinical translation and application of ursolic acid use in cancer. We added a new section titled “patent applications and clinical trials related to ursolic acid use”. This section is added before the conclusion section of the revised manuscript.
- Patent applications and clinical trials related to ursolic acid use
A search of the patent applications (using GooglePatents.com) revealed that, in the past 10 years, 25,638 patent applications have been filed globally pertaining to UA of which 5793 were related to UA use in cancer. In Canada there were 354 applications for UA patents with 140 related to cancer. In the US the numbers were higher with total 1341 applications related to UA and 542 specifically related to UA and cancer. A search of European Union (EU) Clinical Trials Registry and Government of Canada clinical trials registry found no trials registered related to ursolic acid use specifically for cancer treatment or any other use.
A search of ClinicalTrials.gov related to ursolic acid revealed four trials, 3 completed and 1 withdrawn. From the completed clinical trials one (NCT02401113) examined the effect of ursolic acid (derived from Loquat extract) on preventing sarcopenia in 54 adults but no data/results are posted (trial completed Oct 2015). Another trial examined the bioavailability of ursolic acid in 18 healthy male adult participants (NCT04421716) and although the study was completed in April 2021, no results are posted. The third trial (NCT02337933, trial completed Sept 2015) examined the effects of 12 week ursolic acid administration (150mg was given orally once a day) in 24 adult participants with metabolic syndrome [97] and found reduced body weight, BMI, waist circumference, reduced fasting blood glucose levels and improved insulin sensitivity. Around 50% of patients had transient remission of their metabolic syndrome. Patients with insulin resistance and metabolic syndrome have increased risk of developing cancer in general and although this clinical trial is not focused on cancer patients, the data are indirectly relevant to cancer. The reduced metabolic syndrome symptoms with UA use suggest better metabolic control and potential reduced cancer risk. One clinical trial (NCT04403568) involving use of UA (150 mg, twice a day) in the treatment of prostate cancer was posted in May 2020. Unfortunately, this trial was withdrawn due to lack of funding.
These data clearly indicate that although there is evidence from in vitro studies and limited in vivo animal studies of the anticancer potential of ursolic acid, interest from the scientific community, to perform clinical trials is very low. This may be due to lack of strong in vivo animal studies which leads to lack of funding.
One potential limitation of clinical translation of UA is its low bioavailability due to its low water solubility. An innovation that can combat this limitation would be to successfully encapsulate UA into micelles, nanoparticles or liposomes. These encapsulation strategies will increase its water solubility and hopefully will resolve the low bioavailability issue. In one preliminary clinical trial the maximum tolerated dose (MTD) as well as the dose-limiting toxicity (DLT) of ursolic acid liposomes (11-130 mg/m2, administered by a 4h intravenous infusion) was investigated in a group of 63 volunteer subjects. The DLT was found to be between 74 - 130 mg/m2 and mainly consisted of diarrhea and hepatotoxicity, and the MTD was determined to be 98 mg/m2 [98]. The above study [98] together with the study by Xia et al [96] described in section 2.3, (UA nano- liposomes intravenous infusion in humans) and the study by Yang et al [94], described in section 2.3, (UA nano-particles oral administration in rats) are the only studies up to now that attempted to examine dose, bioavailability and toxicity of UA nano-particles. Furthermore, there are no studies examining the effects of such UA nanoparticles against lung cancer (or any other cancer).
Although, few UA derivatives have been found to be more effective than the parent compound, in cell culture studies (studies presented in section 2.3) unfortunately, none of them have been tested in animal or human studies yet. The few studies showing chemo sensitization [73,79,80], and radio sensitization [76] properties of UA are all in cell cultures. One potential future application of UA and UA derivatives is to be used as chemo and/or radio sensitizing agents and hopefully, in vivo studies utilizing lung cancer animal models will be performed in the future to evaluate such a potential.
- For example, the efficacy of UA to different types of lung cancer was not evaluated/compared.
We have added information re the efficacy of UA in different types of lung cancer. This new information is included in section 2.1.
From the studies presented above it is evident that the effects of UA were examined in different lung cancer cells representing different subtypes of the disease. Small cell lung cancer (SCLC) and non-small cell lung cancer (NSCLC) adenocarcinoma, squamous cell carcinoma and large cell carcinoma cell lines were utilized. The effective concentration of UA appears to be in the range of 20-50 µM in the majority of the studies. Unfortunately, there are no available studies that examine and directly compare the UA concentration required for half maximum inhibition (IC50 values) of proliferation of different lung cancer cell lines.
- Besides, patent and clinical trials related to UA were not included in this review too.
We have performed additional search and added information re patent and clinical trials related to UA. This information is included in a new section titled “patent applications and clinical trials related to ursolic acid use”. This section is added before the conclusion section of the revised manuscript.
- Potential limitation and innovation to resolve or promote the clinical translation of the UA can be included too.
We have added information re the potential limitation and innovation to resolve or promote the clinical translation of UA. The new information is included in the new section titled “patent applications and clinical trials related to ursolic acid use”.
- The effect of UA comparing with the currently available lung cancer chemotherapeutic drug shall be evaluated.
We have added information and attempted to compare the effects of UA to the effects seen with currently available lung cancer chemotherapeutic drugs.
Drugs used currently in the treatment of lung cancer include cisplatin, gemcitabine, docetaxel, etoposide, paclitaxel and vinorelbine. Only a few studies have compared the effects of UA to the effects of currently used lung cancer drugs. Kim et al [73] found that UA had similar effects as doxorubicin and veliparib in A549 lung cancer cell. Furthermore, when used in combination UA enhanced the effect of doxorubicin and veliparib. In another study [80] UA was able to overcome paclitaxel resistance in A549 cells.
In vivo animal studies have shown that UA treatment of animals xenografted with lung cancer cells had a similar effect in reducing tumor volume as cyclophosphamide [70], etoposide [73] and doxorubicin [73] and enhanced the effects of etoposide and doxorubicin when used in combination [73]. These studies although very limited provide strong evidence of the anticancer potential of UA.
Reviewer 3 Report
In this review article, the authors revise the role and function of ursolic acid against lung cancer. The manuscript is well written and provides sufficient and relevant information on the topic. I believe there are not any significant issues to discuss.
Author Response
Reviewer 3
Comments and Suggestions for Authors
In this review article, the authors revise the role and function of ursolic acid against lung cancer. The manuscript is well written and provides sufficient and relevant information on the topic. I believe there are not any significant issues to discuss.
Thank you for taking the time to read and review our manuscript. The revisions addressed all comments of the other 2 reviewers and further enhanced the quality of our manuscript.
Round 2
Reviewer 2 Report
Authors have improved the review manuscript based on the suggestions.